# Effectiveness of an Artificial Intelligence Software for Limb Radiographic Fracture Recognition in an Emergency Department

**DOI:** 10.3390/jcm13185575

**Published:** 2024-09-20

**Authors:** Guillaume Herpe, Helena Nelken, Tanguy Vendeuvre, Jeremy Guenezan, Clement Giraud, Olivier Mimoz, Antoine Feydy, Jean-Pierre Tasu, Rémy Guillevin

**Affiliations:** 1Emergency Radiology Unit, University Hospital Center of Poitiers, 86000 Poitiers, France; 2Laboratoire de Mathématiques Appliquées LMA, CNRS UMR 7348, 86021 Poitiers, France; clement.giraud@chu-poitiers.fr; 3Emergency Department, University Hospital Center of Poitiers, 86000 Poitiers, France; tanguy.vendeuvre@chu-poitiers.fr (T.V.); jeremy.guenezan@chu-poitiers.fr (J.G.); olivier.mimoz@chu-poitiers.fr (O.M.); remy.guillevin@chu-poitiers.fr (R.G.); 4Department of Musculoskeletal Imaging, Cochin Hospital, AP-HP, 75014 Paris, France; antoine.feydy@aphp.fr; 5Department of Diagnostic and Interventional Radiology, Poitiers University Hospital, 86000 Poitiers, France; jean-pierre.tasu@chu-poitiers.fr; 6CHU de Poitiers Service de Radiologie, 86000 Poitiers, France

**Keywords:** radiology, fractures, bone, AI, artificial intelligence, emergency department, retrospective study, workflow

## Abstract

**Objectives:** To assess the impact of an Artificial Intelligence (AI) limb bone fracture diagnosis software (AIS) on emergency department (ED) workflow and diagnostic accuracy. **Materials and Methods:** A retrospective study was conducted in two phases—without AIS (Period 1: 1 January 2020–30 June 2020) and with AIS (Period 2: 1 January 2021–30 June 2021). **Results:** Among 3720 patients (1780 in Period 1; 1940 in Period 2), the discrepancy rate decreased by 17% (*p* = 0.04) after AIS implementation. Clinically relevant discrepancies showed no significant change (−1.8%, *p* = 0.99). The mean length of stay in the ED was reduced by 9 min (*p* = 0.03), and expert consultation rates decreased by 1% (*p* = 0.38). **Conclusions:** AIS implementation reduced the overall discrepancy rate and slightly decreased ED length of stay, although its impact on clinically relevant discrepancies remains inconclusive. **Key Point:** After AI software deployment, the rate of radiographic discrepancies decreased by 17% (*p* = 0.04) but this was not clinically relevant (−2%, *p* = 0.99). Length of patient stay in the emergency department decreased by 5% with AI (*p* = 0.03). Bone fracture AI software is effective, but its effectiveness remains to be demonstrated.

## 1. Introduction

Bone fractures have an incidence evaluated between 733 and 4017 per 100,000 patient-years [1]. A missed or delayed diagnosis of fractures on conventional X-rays remains a common problem, ranging from 3% to 10% [2]. Strategies to reduce rates of fracture misdiagnosis are crucial to maintain high standards of patient care, and to limit risks of legal action [3]. Fracture misdiagnosis is a significant concern in clinical settings, particularly in emergency departments where timely and accurate diagnosis is critical. The experience and training of clinicians, especially radiologists, play a crucial role in the accurate identification of fractures. However, several factors contribute to a higher rate of fracture misdiagnosis, including (1) workload and time constraints—in busy emergency departments, clinicians often work under pressure, which can lead to rushed assessments and higher rates of diagnostic errors; (2) complexity of cases—fractures are not always easily identifiable, particularly in areas with complex anatomy or in cases where the fractures are subtle, such as hairline fractures; (3) variability in expertise; and (4) access to expert opinions—while having access to a specialist, such as an orthopedic surgeon or an experienced radiologist, can significantly reduce diagnostic errors, such access is not always feasible, especially in remote or under-resourced areas.

Given the variability in clinician expertise and the challenges of accessing expert opinions, Artificial Intelligence (AI) has emerged as a valuable tool in assisting with fracture diagnosis. AI-powered tools can (1) support clinicians in diagnosis by identifying fractures with high accuracy, acting as a second set of eyes for clinicians; (2) reduce diagnostic discrepancies, especially in subtle or complex cases; and (3) bridge the expertise gap. In situations where access to a radiology expert is limited, AI can provide decision support, ensuring that even less-experienced clinicians are empowered to make more accurate diagnoses.

These AI tools are particularly valuable in emergency radiology, where speed and accuracy are crucial for optimal patient outcomes [4]. For instance, in some emergency departments, a radiologist on duty is not available 24/7 for radiography reading [5,6].

Artificial Intelligence (AI) including machine learning and deep learning has been used to enable algorithms to learn from data, iteratively improving their own performance without the need for explicit programming. AI has rapidly been put into use in imaging, as a decision aid, as a screening tool, or as a second-reader support for radiologists [3,7].

In bone fracture detection and classification, recent reviews and meta-analyses have reported a high accuracy for AI, with a sensitivity of 92% and 91% and a specificity of 91% on internal and external validation, respectively [8,9,10,11]. In a recent meta-analysis of 42 studies, including 37 on radiography, diagnostic performance was comparable for AI and clinicians [11].

In recent years, AI software has been designed as a diagnostic aid with the aim of improving workflow through screening or prioritizing images on worklists and highlighting regions of interest for the clinician [12]. AI could also improve diagnostic certainty by acting as a “second reader” for clinicians when no radiologist is present. However, these advantages remain theoretical, and sometimes commercial, mostly being based on efficacy rather than effectiveness [13].

Given the increasing integration of AI technologies in clinical settings, particularly in emergency departments (EDs), there is a crucial need to evaluate their practical effectiveness. Emergency departments are at the forefront of acute care, where rapid and accurate diagnosis is paramount. This study therefore aims to assess the real-world impact of an AI-based diagnostic tool within this critical environment.

## 2. Materials and Methods

A consecutive before–after pragmatic retrospective study was conducted in a single French public University Hospital during two periods of time—one before and one after introducing AIS; from 1 January 2020 at 00:01 am to 30 June 2020 at 11:59 pm, called Period 1, no AI was available. During the second period, between 1 January 2021 at 00:01 am and 30 June 2021 at 11:59 pm, called Period 2, the commercially available Bone View (V1., Gleamer, Paris, France) was available for clinicians in the ED.

The data were collected from Poitiers University Hospital, an academic level I trauma center in a 200,000 people referral area with 75,000 visits per year.

This retrospective study was approved by the institutional review board of Comité Ethique et Recherche en Imagerie Médicale (protocol code: CRM-2204-263 and approval date: 23 April 2023). Written informed consent was waived due to retrospective anonymized data collection. The study flowchart is given in Figure 1.

Two months before Period 2, the AIS was set up (1 November 2020) and a dedicated training session was given to physicians and radiologists. The training consisted of 2 sessions of 1 h (provided by G.H) looking at clinical cases including AIS solution. The AIS validation method and the code were provided in a previously published study [14] and in the Appendix A. This version of the AIS was only able to analyze the appendicular skeleton (lower and upper extremities). Spine and pelvic bones were excluded from the analysis.

AIS highlighted a potential fracture in a rectangular box and results were available in less than 1 min on both the PACS system for radiologists (Change Healthcare Radiology v14, Mc Kesson, Vancouver, BC, Canada) and on the emergency physicians’ radiographic viewers.

In total, 22 senior emergency physicians and 7 senior radiologists with more than 10 years of experience participated during the 2 periods of the study. ED physicians and radiologists along with their years of experience are listed in the Appendix A. All participants have no conflicts of interest.

### 2.1. Inclusion and Exclusion Criteria

All consecutive patients with limb bone fracture suspicion visiting the ED during Period 1 and Period 2 were included retrospectively from the hospital electronic medical records (Telemaque V14, Poitiers, France).

Inclusion criteria were age (17 years or older), referred to the ED after a recent trauma of less than 72 h, and having undergone a limb radiograph (shoulder, elbow, arm, wrist, hand, hip, knee, leg, ankle, or foot).

To avoid the influence of potential CT scan read back, patients having undergone a CT scan after radiographs were excluded. We also excluded patients for whom radiographs were not read by a radiologist and in-patients for whom radiographs could be analyzed by different physicians in different specialties during hospitalization.

### 2.2. Gold Standard

The radiologist’s reading was considered as the gold standard. The reading was performed without or with the help of AIS, according to the period. In the cases of discrepancy with an ED physician, a second radiologist reviewed the case to set a final diagnosis. The radiologist’s reading (unique or in consensus) was considered as the gold standard.

### 2.3. Evaluation Criteria

To assess the effectiveness of AIS, the following parameters were recorded:Discrepancy rate of limb fracture diagnosis between radiologists and emergency physicians. Discrepancies were defined as differences in the presence or lack of fracture between the final discharge summary performed by the emergency physicians and the radiologist’s report. To avoid any mutual influence, emergency physicians established their final diagnosis blinded to the radiologist’s report available 24 to 72 h after patient discharge. On the contrary, to alert the clinician to a possible discrepancy, radiologists had knowledge of the final diagnosis of emergency physicians during their readings.Length of stay in the ED was defined as the interval between arrival time at the ED and discharge time from the ED and was expressed in minutes and was recorded from the EHR.Number of clinician experts’ opinions requested whatever the specialty, e.g., orthopedics, radiologists, or resuscitation physicians.Any changes in patient management induced by the radiologist’s report including changes in treatment (new immobilization with a splint or a plaster, withdrawal of immobilization in the absence of fracture) or follow-up (for instance, new orthopedic appointment) were recorded. All modifications were assessed retrospectively, in consensus by an orthopedic surgeon (TV), an emergency radiologist (GH), and an ED physician (JG), all with more than 10 years of experience, and were defined as being clinically relevant or not. For example, discrepancy was not considered clinically relevant if an elbow fracture was missed but the patient was correctly immobilized with a splint.

### 2.4. Statistical Analysis

Continuous variables are presented as means and standard deviations (SDs) or medians with ranges, depending on whether they have a normal distribution or not. Categorical variables are presented as numbers with rate. The statistical tests used were Student’s *t*-test, the Wilcoxon–Mann–Whitney test and the test of difference in proportions, where its effect size is defined as [15] $h=-\phi_{2}$, where
$=2\;arcsine\left(\sqrt{P_{i}}\right)$. An h near 0.2 is a small effect, an h near 0.5 is a medium effect, and an h near 0.8 is a large effect.

We trained a logistic regression model that estimates the probability of discrepancy, presenting the following features: injury location, use of AIS, age, orthopedic appointment within a month, and orthopedic second opinion. The forest plot of this model is displayed in Figure 2 [16,17,18].

No imputation was made for missing data. Statistical analysis was performed using R software, version 4.0.4.

## 3. Results

Among the 41,571 patients visiting the ED during the two study periods (19,991 patients during P1; 21,580 during P2), 16,418 (39%) underwent radiography (7984 [40%] during P1; 8434 [39%] during P2). In total, 4707 (11%) patients fulfilled the inclusion criteria and were therefore included. Of these 4707 patients, 987 (21%) were secondarily excluded because they were later hospitalized (*n* = 864, 18%) or had a CT scan (246, 5%). Finally, 3720 patients (1780 during Period 1 and 1940 during Period 2) were included in the final analysis. 

The clinical characteristics of patients were similar in the two study periods (Table 1). In contrast, there were small differences in the site of the suspected injury between the two periods. There were three times as many shoulder radiographs in Period 2 (Table 1). Other differences were not significant.

With the use of AIS, discrepancy rates decreased by 17% from 6.6% in Period 1 to 5.1% in Period 2 (*p* = 0.04) (Table 2). The reduction was significant only for the arm radiographs (Table 3). Discrepancies were more frequent for wrist and elbow (*p* = 0.003) compared to other locations. The discrepancy rate was not associated with either patient age (OR = 1.14, range 0.80–1.62; *p* = 0.45) or orthopedic opinion request (OR = 1.26, range 0.72–2.05, *p* = 0.39) (Figure 2).

Mean length of stay in the ED decreased from 190 min (±132 min) for Period 1 to 181 (±137) for Period 2 (*p* = 0.03) (Table 2).

For the two same periods, patient length of stay in the ED did not significantly change from 306 min (±125) to 305 min (±129) (*p* = 0.40).

Expert referral rate did not statistically differ between Period 1 (7.1%) and Period 2 (6.3%) (*p* = 0.40). In the month following patient ED discharge, 810 patients (22%) were consulted by an orthopedic surgeon, with no difference between the two study periods (377 (21) % in Period 1 and 433 (22.3%) in Period 2) (*p* = 0.42).

In total, 118 of 216 discrepancies were clinically relevant with an impact on the patient management (55%). There were no differences in the levels of discrepancy for the two periods (65/118 patients [55%] in Period 1 and 53/98 [54%] patients in Period 2) (*p* = 0.99).

## 4. Discussion

Our study aimed to assess how the implementation of AI technology for limb fracture diagnosis affects patient care in the ED. Our results show that AIS has no major effect on clinically relevant discrepancies, despite reducing the rate of total discrepancies by 17%. The need for expert advice remains unchanged and the length of stay in the ED is only minimally affected after AIS setup.

To our knowledge, this study is the first to evaluate the impacts of AIS in real-life conditions in an ED. It should be highlighted that contrary to previous studies, neither the population set nor the readers were selected, and the entire ED physician and emergency radiologist teams participated in the two periods of the study [14,19,20]. Experiences of the teams are provided within the Appendix A.

Our findings are similar to previously reported distributions of fractures among age and anatomical location.

Three points require discussion:

First, the decrease in mean length of stay in the ED is obviously small (9 min on average per patient). However, the number of emergency admissions should be keep in mind; for instance, when there are 100 visits for a potential limb injury per day, this represents a theoretical gain of 15 h. It should be noted that no changes were made to the ED patient workflow after the introduction of AI and it is likely that the addition of triage [21], prioritization [22] and/or care coordination [23] AI functions could have had a greater impact on this time saving.

Second, the overall reduction in discrepancies of 17% could be considered as interesting and regarded as a significant quality improvement for the patient. Indeed, while the first physician’s interpretation facilitates prompt management, reading errors can potentially place the patient at an unnecessary risk of adverse outcomes and the physician at risk of litigation. Discrepancy rates may vary depending on clinicians’ expertise. Several studies with controlled data sets have investigated differences in radiographic interpretation between emergency physicians and radiologists, yielding varying levels of agreement ranging from 52% to 99% [24,25,26,27].

In our study, AIS has no significant effect on clinically relevant discrepancies. This is in line with a study by Tranovich et al. [5], which found 1044 radiographic discrepancies among 16,111 physician interpretations, with only 0.01% for relevant limb fracture. One should outline that the clinical examination′s performance and the frequent expert opinion recourse might act as additional filters to ensure patient safety, preventing treatment based solely on radiographic findings.

Thirdly, the requirement for specialist consultation was not affected by AIS. This is probably due to the necessity for advice on both diagnosis and treatment.

Our study has some limitations, as follows:

First, we did not stratify patient workflow regarding emergency physician experience. This stratification could be interesting, particularly during Period 1. For Period 2, this stratification is less relevant since standalone diagnostic performances of the AI do not rely on ED physicians [14].

Second, the retrospective design could be considered as a weakness. For instance, the number of specialist opinion requests was recorded retrospectively from electronic medical reports, and informal calls were likely omitted. However, without any information being given to the physicians and radiologist on the study, the risk of individual behavior modifications in response to their awareness of being observed was avoided [28].

Third, this is a monocentric study with all the limits due to this design. Our ED might have a specific workflow and the University Hospital cannot reflect the general situation of a healthcare system.

Fourth, we wish to underline that our study was performed during 2020 and 2021, and therefore used the 2021 version of the Boneview algorithm, with the exclusion of spine, ribs, and pelvis. As every release of AIS increases performance and covered anatomical locations, the new version of AIS might have a better performance. This could therefore underestimate the effectiveness of an AIS on the overall common indication of post traumatic radiographs.

Fifth, the observed reduction in hospital length of stay can have different non-controlled causes; several methods have been developed to target a shorter length of stay such as better care coordination, a specific discharge planning program, dedicated case management, a redesigned staffing model, or specialized units for high-risk populations [29]. All of these and other potential management changes in the ED were not controlled for in this study. However, few structural or organizational changes occurred in the emergency department between the two periods. Medical and nursing staff changed only marginally. The number of emergency room visits increased over the period.

Lastly, to assess the financial impact of AI software on a medical institution, a further medico-economic analysis should be conducted.

Despite these limitations, the findings suggest that AIS can contribute to reducing diagnostic discrepancies in busy ED settings, potentially improving patient flow and reducing time-to-diagnosis. However, the lack of significant impact on clinically relevant discrepancies indicates that AIS should be seen as an adjunct to, rather than a replacement for, expert clinical judgment.

## 5. Conclusions

Clinically relevant discrepancies (those that could have altered patient management) remained stable at around 55% of all discrepancies, despite the overall reduction in discrepancies. This suggests that while AI helped reduce some discrepancies, it did not significantly change the proportion that had clinical relevance, i.e., those that would have added clinical value if detected earlier or differently. Length of stay in the emergency department is only slightly affected by the AIS. These findings underline the importance of distinguishing between general effectiveness (reduced errors) and clinical effectiveness (improved patient outcomes). Further prospective multicentric studies could explore deeper integration to achieve effectiveness rather than efficacy.

## Figures and Tables

**Figure 1 jcm-13-05575-f001:**
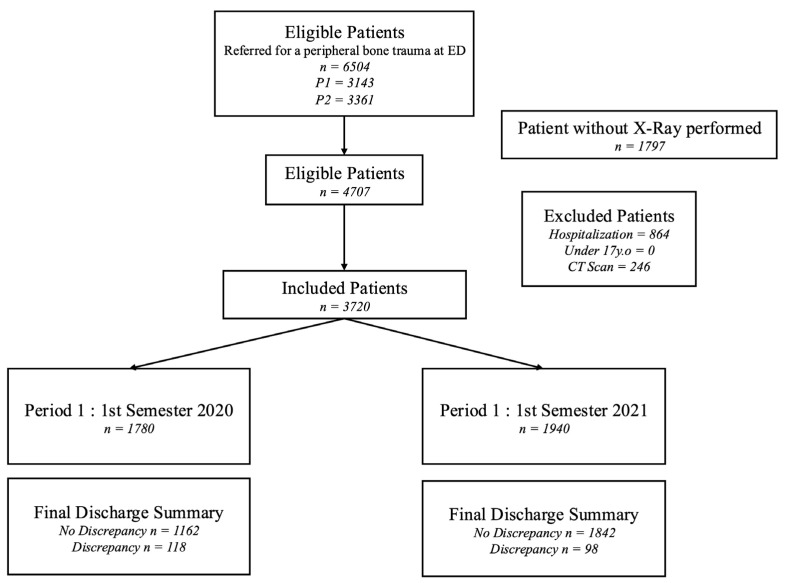
Flowchart of the study.

**Figure 2 jcm-13-05575-f002:**
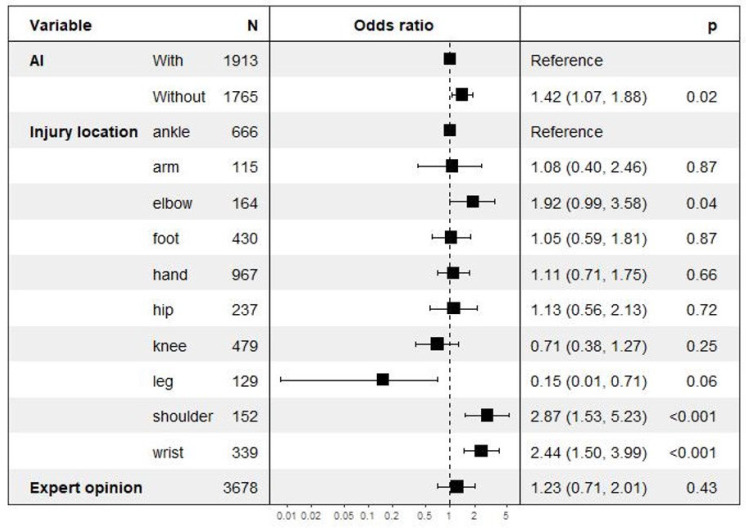
Multivariate risk analysis regarding discrepancies focusing on AI deployment along with fracture location and expert opinion.

**Table 1 jcm-13-05575-t001:** The demographic characteristics of the study along with the difference among the two subsets and the statistical significant after multiple testing corrections. ** significant *p* value after multiple testing corrections. SD: standard deviation; AIS: Artificial Intelligence limb bone fracture diagnosis software.

	Without AIS *N* = 1780	With AIS *N* = 1940	*p* Value	Effect Size
Age (mean ± SD, in years)	42 ± 20	42 ± 21	0.82	
Sex (number, (%))				
Male	974 (54.7)	1102 (56.8)	0.21	0
Injury location (number, (%))				
Hand	449 (25.2)	518 (26.7)	0.3231	0
Ankle	335 (18.8)	331 (17.1)	0.1755	0
Arm	53 (3.0)	62 (3.2)	0.7722	0
Foot	207 (11.6)	223 (11.5)	0.9389	
Hip	114 (6.4)	123 (6.3)	0.9896	0
Knee	244 (13.7)	235 (12.1)	0.1611	0
Leg	69 (3.9)	60 (3.1)	0.2243	0
Wrist	172 (9.7)	167 (8.6)	0.2894	0
Shoulder	36 (2.0)	116 (6.0)	<0.0001 **	−0.2
Elbow	86 (4.8)	78 (4.0)	0.2612	0

**Table 2 jcm-13-05575-t002:** The difference before and after AIS deployment on the primary and secondary criteria along with the effect size (An h near 0.2 is a small effect, an h near 0.5 is a medium effect, and an h near 0.8 is a large effect). SD: standard deviation. AIS: Artificial Intelligence limb bone fracture diagnosis software.

	Before AIS (*N* = 1780)	With AIS (*n* = 1940)	*p* Value	Effect Size
Discrepancy (number, %)	118 (6.6)	98 (5.1)	0.04	0.1
Clinically relevant discrepancies (number, (%))	65/118 (55)	53/98 (54)	0.99	0
Length of stay (mean ± SD in minutes)	190 ± 132	181 ± 137	0.03	-
Orthopedic opinion referral rate (number, %)	126 (7.1)	123 (6.3)	0.40	0

**Table 3 jcm-13-05575-t003:** Distribution of discrepancies according to the location of the trauma on the overall study period; before and after the trauma along with the effect size. * significant *p* value after multiple testing corrections. AIS: AI limb bone fracture diagnosis software.

Injury Location (Fracture Rate, %)	Overall	Before AIS	With AIS	*p* Value	Effect Size
Foot	22/430 (5.12%)	11/207 (5.31%)	11/223 (4.93%)	1	0
Ankle	33/666 (4.95%)	22/335 (6.57%)	11/331 (3.32%)	0.08	0.2
Elbow	15/164 (9.15%)	11/86 (12.79%)	4/78 (5.13%)	0.15	0.3
Hip	13/237 (5.49%)	9/114 (7.89%)	4/123 (3.25%)	0.19	0.2
Hand	52/967 (5.38%)	24/449 (5.35%)	28/518 (5.41%)	1	0
Wrist	38/339 (11.21%)	20/172 (11.63%)	18/167 (10.78%)	0.93	0
Arm	6/115 (5.22%)	6/53 (11.32%)	0/62 (0%)	0.02 *	0.7
Knee	17/479 (3.55%)	6/244 (2.46%)	11/235 (4.68%)	0.28	−0.1
Shoulder	18/152 (11.84%)	8/36 (22.22%)	10/116 (8.62%)	0.055	0.4
Clavicle	0/28 (0%)	0/8 (0%)	0/20 (0%)	-	-
Leg	1/129 (0.78%)	1/69 (1.45%)	0/60 (0%)	1	0.2
Calcaneus	0/8 (0%)	0/3 (0%)	0/5 (0%)	-	-
Upper limb	0/5 (0%)	0/4 (0%)	0/1 (0%)	-	-
Bone	1/1 (100%)	0/0 (-%)	1/1 (100%)	-	-
Overall	216/3720 (5.81%)	118/1780 (6.63%)	98/1940 (5.05%)	0.04 *	0.1

## Data Availability

The data presented in this study are available on request from the corresponding author.

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
