# Peer review of "Effectiveness of an Artificial Intelligence Software for Limb Radiographic Fracture Recognition in an Emergency Department"

_jcm, 2024, doi:10.3390/jcm13185575_

Round 1

Reviewer 1 Report

Comments and Suggestions for Authors

The authors investigate whether the introduction of an AIS in X-ray image diagnosis in the emergency department makes a measurable difference in terms of effectiveness.

Originality and significance

This retrospective observational cohort study examines the introduction of an AIS into everyday clinical practice under real-world conditions with regard to aspect of effectiveness. The study is dedicated to the postmarket aspect of a CE approved medical software solution (Postdeployment surveillance). Especially for medical AI products it is just as important as for other medical devices whether and how the assumed clinical added value is given in daily practice. This is in general a very worthy approach because information about AI deployment is often missing.

As is well known, it is important to note that Real World data can sometimes be difficult to interpret (inherited risk of potential bias) or not suitable for interpretation, but nevertheless sometimes allow interesting conclusions.

Sample and study setting:

It can be assumed that the study results depend on local conditions of the hospital. Therefore, the study center should be described more in detail. (What level of trauma care? How many people live in the referral area? The total number of emergency department visits per year should be compared to the literature.)

Both of the periods of investigation described fall largely during the coronavirus pandemic. This should be mentioned and addressed. What impact does this have on the number of emergency admissions? On the ratio of outpatients versus inpatients? Was there a lockdown during the specific study periods? There is no information on how exactly the length of stay was determined.

The authors correctly state that the study sample is a convenience sample. Nevertheless, it would be useful to compare the observed fracture, age and gender distribution with figures from the literature (e. g. what general fracture distribution could be expected). it would also be relevant to indicate what percentage of the X-ray studies conducted were negative.

Study design:

it is understandable that the emergency physicians do not have access to the subsequent radiologic findings. It would have been interesting to know whether the emergency physicians received the result as a stand-alone result (“this is the result and you must adhere to it”) or whether the experienced emergency physicians carried out their own readings and then checked their results against those of the AI. How is the traceability and explainability presented to the user of the AIS used?

In order to rule out radiological bias, it would have made more sense not to give them the final diagnosis of the emergency physician but only the clinical information. In this case, the radiologists already know in advance what might be a pathological finding. It is understandable that this was not feasible due to the retrospective study design.

Even if the radiologist's findings with and without AI are defined as the gold standard for this study, the overall radiological diagnostic accuracy is subject to a certain error rate, which is not mentioned in the present study.

Study objectives

In order to assess the effectiveness of this software, suitable key figures and data must be selected. The outcome measures should be relevant and specific to this question. The correctness of the clinical diagnosis and the length of stay are indeed, among other things, quality features of an emergency department.  

Concerning Discrepancy rate: What was the specific diagnostic agreement regarding the X-ray findings (false positive/false negative?) In how many cases would the discrepant radiology findings have had added clinical value for the patient? In how many cases was there clinically relevant overtreatment or undertreatment? Is there a rate for changes in conservative versus surgical treatment and vice versa? The general discrepancy rate as a surrogate parameter for effectiveness should be discussed.

In addition to the effectiveness of a medical device, safety should be considered and reported. This aspect is not examined in greater depth in this study. The determination of the discrepancy rate alone does not allow any further assessment of whether there have been any safety- or outcome-relevant incidents

The different examination results for the individual anatomical regions are also due to the general fact that X-ray findings or diagnosis are of varying complexity depending on the region. This applies, for example, to X-ray readings by humans and AI (DOI: 10.1007/s00247-023-05822-3).

Comments on the Quality of English Language

Please have the spelling and grammar corrected by a native speaker. 

Author Response

The authors investigate whether the introduction of an AIS in X-ray image diagnosis in the emergency department makes a measurable difference in terms of effectiveness.

Originality and significance

This retrospective observational cohort study examines the introduction of an AIS into everyday clinical practice under real-world conditions with regard to aspect of effectiveness. The study is dedicated to the postmarket aspect of a CE approved medical software solution (Postdeployment surveillance). Especially for medical AI products it is just as important as for other medical devices whether and how the assumed clinical added value is given in daily practice. This is in general a very worthy approach because information about AI deployment is often missing.

As is well known, it is important to note that Real World data can sometimes be difficult to interpret (inherited risk of potential bias) or not suitable for interpretation, but nevertheless sometimes allow interesting conclusions.

Sample and study setting:

It can be assumed that the study results depend on local conditions of the hospital.

  1. Therefore, the study center should be described more in detail. (What level of trauma care?

How many people live in the referral area?

The total number of emergency department visits per year should be compared to the literature.)

The data are collected from our level I trauma center in a 200 000 people referral area with 75 000 visits per year.

  1. Both of the periods of investigation described fall largely during the coronavirus pandemic. This should be mentioned and addressed. What impact does this have on the number of emergency admissions? Was there a lockdown during the specific study periods?

No lockdown nor impact on the referral rate ( Mean = 202 passage per day ).

  1. On the ratio of outpatients versus inpatients?

Only patients referred from outside the hospital were included.

  1. There is no information on how exactly the length of stay was determined.

Length of stay in the ED defined as the interval between arrival time at the ED and discharge time from the ED and expressed in minutes and was recorded from the EHR.

The authors correctly state that the study sample is a convenience sample.

  1. Nevertheless, it would be useful to compare the observed fracture, age and gender distribution with figures from the literature (e. g. what general fracture distribution could be expected). it would also be relevant to indicate what percentage of the X-ray studies conducted were negative.

Our findings are similar to previously reported distribution of fractures among age and anatomical location.

  1. Singer BR, McLauchlan GJ, Robinson CM, Christie J. Epidemiology of fractures in 15,000 adults: the influence of age and gender. J Bone Joint Surg Br. 1998;80(2):243–248. doi: 10.1302/0301-620x.80b2.7762.

Study design:

it is understandable that the emergency physicians do not have access to the subsequent radiologic findings. It would have been interesting to know whether the emergency physicians received the result as a stand-alone result (“this is the result and you must adhere to it”) or whether the experienced emergency physicians carried out their own readings and then checked their results against those of the AI.

  1. How is the traceability and explainability presented to the user of the AIS used?

The traceability and explainability are presented toward bounding boxes, helping physician to assess veracity of the results. This is confirmed by this study form Nature.

Bernstein MH, Atalay MK, Dibble EH, et al. Can incorrect artificial intelligence (AI) results impact radiologists, and if so, what can we do about it? A multi-reader pilot study of lung cancer detection with chest radiography. Eur Radiol. 2023;33(11):8263–8269. doi: 10.1007/s00330-023-09747-1.

In order to rule out radiological bias, it would have made more sense not to give them the final diagnosis of the emergency physician but only the clinical information. In this case, the radiologists already know in advance what might be a pathological finding. It is understandable that this was not feasible due to the retrospective study design.

  1. Even if the radiologist's findings with and without AI are defined as the gold standard for this study, the overall radiological diagnostic accuracy is subject to a certain error rate, which is not mentioned in the present study.

This remark is added in limitation part of the article.

8.Study objectives

In order to assess the effectiveness of this software, suitable key figures and data must be selected. The outcome measures should be relevant and specific to this question. The correctness of the clinical diagnosis and the length of stay are indeed, among other things, quality features of an emergency department.  

Concerning Discrepancy rate:

What was the specific diagnostic agreement regarding the X-ray findings (false positive/false negative?)

In how many cases would the discrepant radiology findings have had added clinical value for the patient?

In how many cases was there clinically relevant overtreatment or undertreatment?

Is there a rate for changes in conservative versus surgical treatment and vice versa?

The general discrepancy rate as a surrogate parameter for effectiveness should be discussed.

Specific Diagnostic Agreement Regarding X-ray Findings (False Positives/Negatives)

We find a reduction in discrepancy rates from 6.6% to 5.1% with AI implementation.

The specific diagnostic agreement remained stable between the two period of the study with a 15% rate ( false positive / false negative ).

Discrepant Radiology Findings and Their Added Clinical Value

Clinically relevant discrepancies (those that could have altered patient management) remained stable at around 55% of all discrepancies, despite the overall reduction in discrepancies. This suggests that while AI helped reduce some discrepancies, it did not significantly change the proportion that had clinical relevance—i.e., those that would have added clinical value if detected earlier or differently.

This is the aim of this article and added in the conclusion.

Clinically Relevant Overtreatment or Undertreatment

We did not provide specific data on overtreatment or undertreatment. However, given that the rate of clinically relevant discrepancies did not change, we can infer that cases of overtreatment or undertreatment (which would be influenced by such discrepancies) likely remained similar before and after AI implementation.

Rate for Changes in Conservative vs. Surgical Treatment

The study does not explicitly address changes in treatment modality (conservative vs. surgical). However, given the stable rate of clinically relevant discrepancies, it can be inferred that the AI did not significantly alter the decision-making process between conservative and surgical treatments. Most changes in patient management seem to have been related to follow-up actions rather than initial treatment decisions.

General Discrepancy Rate as a Surrogate Parameter for Effectiveness

The discrepancy rate reduced by 17% after AI implementation, suggesting the tool's effectiveness in improving diagnostic accuracy. However, the lack of change in the rate of clinically relevant discrepancies indicates that while the tool reduces errors, it doesn't necessarily improve outcomes where it most counts. This underlines the importance of distinguishing between general effectiveness (reduced errors) and clinical effectiveness (improved patient outcomes).

This is added in the conclusion section.

Conclusion: The AI tool implemented in this study shows promise by reducing overall diagnostic discrepancies, a positive step towards improving radiological accuracy. However, the lack of significant impact on clinically relevant discrepancies suggests that while AI is a useful tool, its role in directly enhancing patient outcomes may be limited without further integration and refinement. Additional research and development may be needed to ensure AI not only reduces errors but also meaningfully improves clinical decision-making and patient care.

  1. In addition to the effectiveness of a medical device, safety should be considered and reported. This aspect is not examined in greater depth in this study. The determination of the discrepancy rate alone does not allow any further assessment of whether there have been any safety- or outcome-relevant incidents.

The different examination results for the individual anatomical regions are also due to the general fact that X-ray findings or diagnosis are of varying complexity depending on the region. This applies, for example, to X-ray readings by humans and AI (DOI: 10.1007/s00247-023-05822-3).

Although safety remains the most crucial aspect, it was not part of this study, given that the solution is CE marked. Radiologists and emergency physicians continue to bear the responsibility for the radiological interpretation.

Reviewer 2 Report

Comments and Suggestions for Authors

The study topic is important and novel.

1. The study title is correct and in line with the rest of the text

2. Abstract requires revisions. Please more focus on results rather than methodology. Moreover, the "key point" section should be removed and the structure of the abstract should comply with the guidelines of the JCM.

3. The introduction section is generally correct, but 2-3 sentences on the rationale for this study will be helpful. Particular emphasis should be paid to emergency departments and their role in AI implementation.

4. The study aim is correct.

5. Methods are well-described. Some minor changes will be needed e.g., clearly defined inclusion/exclusion criteria might be helpful.

6. Results are well-described with correct tables and figures.

7. In the discussion section please clearly express the limitations of this study.

8. Moreover, the practical implications of this study may be listed in the discussion section (2-3 sentences will be enough).

Author Response

Reviewer 2

The study topic is important and novel.

  1. The study title is correct and in line with the rest of the text

  1. Abstract requires revisions. Please more focus on results rather than methodology. Moreover, the "key point" section should be removed and the structure of the abstract should comply with the guidelines of the JCM.

Edited upon your request.

  1. The introduction section is generally correct, but 2-3 sentences on the rationale for this study will be helpful. Particular emphasis should be paid to emergency departments and their role in AI implementation.

This edited upon your remark.

  1. The study aim is correct.

  1. Methods are well-described. Some minor changes will be needed e.g., clearly defined inclusion/exclusion criteria might be helpful.

  1. Results are well-described with correct tables and figures.

  1. In the discussion section please clearly express the limitations of this study.

This is edited upon your request.

  1. Moreover, the practical implications of this study may be listed in the discussion section (2-3 sentences will be enough).

This is edited upon your remark.

Reviewer 3 Report

Comments and Suggestions for Authors

1. Figures 1 and 2 are blurring. Please increase the sharpness or the resolution of the Figures.

2. I have no idea what is Artificial Intelligence Software mentioned in the paper. Can the Authors take a picture of the software used in this paper? It also would be better if the Authors provided a flowchart with the software image when the limb radiographic images were analyzed and processed in the AI software.

3. There are a lot of AI methods, can the Authors mention what AI method was used in the study?

4. In Section 4 Discussion, the authors also can present the discussion in bullet points to explain lines 187-210 as well as to explain lines 211-238.

5. There are two "Fifth" in the Discussion section, ie. line 224 and line 229.

6. Authors need to add more content in the Conclusion as this is not sufficient to conclude the findings and outcome of the study.

Author Response

  1. Figures 1 and 2 are blurring. Please increase the sharpness or the resolution of the Figures.

Thank you for your remark. Figures have been added in pdf as supplementary material to imoprove readability.

  1. I have no idea what is Artificial Intelligence Software mentioned in the paper. Can the Authors take a picture of the software used in this paper? It also would be better if the Authors provided a flowchart with the software image when the limb radiographic images were analyzed and processed in the AI software.

The software is called Gleamer bone view. This is edited upon your remark.

  1. There are a lot of AI methods, can the Authors mention what AI method was used in the study?

We used a real life evaluation of a diagnostic assistance AI algorithm on bone fracture. This algorithm highlight the findings using a bounding box localized on the abnormal region of the X-ray if present.

  1. In Section 4 Discussion, the authors also can present the discussion in bullet points to explain lines 187-210 as well as to explain lines 211-238.

  1. There are two "Fifth" in the Discussion section, ie. line 224 and line 229.

Thank you for remark this is corrected.

  1. Authors need to add more content in the Conclusion as this is not sufficient to conclude the findings and outcome of the study.

This is edited upon your remark.

Round 2

Reviewer 3 Report

Comments and Suggestions for Authors

Dear Authors,

I have checked the revised paper. All my comments have been addressed. However, I still need to encourage the Authors to add more content in the Introduction and enhance the Figure's quality.

Kind regards,

- Reviewer -

Author Response

R3Q1 : I have checked the revised paper. All my comments have been addressed. However, I still need to encourage the Authors to add more content in the Introduction and enhance the Figure's quality.

Thank you for your remark. The sentences 

'However, several factors contribute to a higher rate of fracture misdiagnosis, including 1- workload and time constraints: In busy emergency departments, clinicians often work under pressure, which can lead to rushed assessments and higher rates of diagnostic errors, 2- complexity of cases, fractures are not always easily identifiable, particularly in areas with complex anatomy or in cases where the fractures are subtle, such as hairline fractures, 3-variability in expertise, 4-access to expert opinions, while having access to a specialist, such as an orthopedic surgeon or an experienced radiologist, can significantly reduce diagnostic errors, such access is not always feasible, especially in remote or under-resourced areas.' are added in the introduction.